# Perception and reasons for the choice of informal provider among women receiving antenatal care services from traditional birth attendants in rural communities of Lagos state, Nigeria

Esther O. Oluwole[1]*, Tolulope E. Oluwadumila[1], Ifeoma P. Okafor[1], Abiola O. Temitayo-Oboh[2]

1 Department of Community Health and Primary Care, College of Medicine, University of Lagos, Lagos, Lagos State, Nigeria, 2 Department of Community Medicine and Primary Care, Federal Medical Centre, Abeokuta, Ogun State, Nigeria

☯ These authors contributed equally to this work.

* oluester2005@yahoo.com

**Data Availability Statement:** All relevant data are within the manuscript and its Supporting information files.

## Abstract

### Background

Unskilled delivery, particularly the use of traditional birth attendants, is a major threat to reducing maternal mortality in Africa. Despite the associated risks, there is insufficient evidence on the major reasons why pregnant women in Nigeria continue to use traditional birth attendant (TBA) services, especially in rural areas. This study, therefore, assessed the perception, reasons for use, and utilization of only TBA services in current pregnancy among rural-dwelling pregnant women in Lagos, Nigeria.

### Methods

A descriptive cross-sectional study was conducted among 347 pregnant women recruited from traditional birth attendant facilities at Ikorodu Local Government Area, Lagos. Data were collected with an interviewer-administered questionnaire and analyzed using SPSS version 25. Bivariate and multivariate analyses were conducted with a significance level set at p<0.05. The outcome measures included perception, utilization of only TBA services in current pregnancy, and reasons for use. Positive perception refers to positive thought, belief, or opinion held by the participants towards the TBA.

### Results

All the respondents had a positive perception of TBAs, majority (70.3%) utilized TBA only while 29.7% combined TBA and healthcare facilities services in the current pregnancy. Recommendations from previous users (81.6%), welcoming and hospitable staff (77.2%), perceived spiritual protection (75.2%), and past use (68.6%), were some of the reasons cited by the respondents for utilizing only TBA services. Predictors of utilizing only TBA services

**Funding:** The author(s) received no specific funding for this work.

**Competing interests:** The authors have declared that no competing interests exist.

were respondents' level of education, those with secondary (aOR = 0.261; 95% CI; 0.108–0.629) and tertiary (aOR = 0.352; 95% CI; 0.162–0.769) had lower utilization while the lack of health insurance coverage (aOR = 3.017; 95% CI; 1.476–6.166) were associated with higher utilization of TBAs.

## Conclusion

Respondents in this study had a positive perception of TBA services. Continuous training and an effective monitoring system of TBAs by the government and other healthcare stakeholders to improve women's birthing experiences is recommended.

## Introduction

Maternal mortality is disproportionately high in developing countries [1]. It contributes more than any other cause to the differences in average life expectancy between women and men [2]. Unskilled home delivery, particularly the use of traditional birth attendants (TBA), is one of the largest contributors to maternal mortality in Africa [3]. Approximately three-quarters of maternal deaths occur from complications during labor, delivery, and the first 24 hours postpartum [4]. The practices of TBAs harm maternal and child health. They may result in complications and untimely deaths since TBA are not usually trained to predict, risk-stratify, or appropriately manage obstetric emergencies [5, 6]. A traditional birth attendant (TBAs) is a person who assists the mother during childbirth and who initially acquired her skills by delivering babies herself or by working with other TBA [7]. Deliveries conducted by TBA are associated with four times higher morbidity and mortality when compared with deliveries supervised by midwives and other health professionals [5]. This is because the TBA do not possess biomedical training and resources, further contributing to maternal and neonatal mortality [8]. Thus, skilled attendance at delivery is a critical intervention toward reducing maternal mortality [4]. Skilled birth attendance requires two major components: a skilled birth attendant, and an enabling environment that includes drugs and equipment, a functional referral system, and supportive policies [9].

The major reasons for the preference for TBA services by pregnant women, and the factors influencing this choice are not well understood [10]. Women perceive TBA positively and utilize them due to accessibility, affordability, and previous positive experiences with them. The community has a high level of trust in them and considers them essential in providing maternal health services [11, 12]. Moreover, TBAshase long been considered respectable birth attendants for women during childbirth in Africa and as major caregivers, especially in rural areas [13, 14], and this is no exception in Nigeria.

Births assisted by TBAs are common in Nigeria and prevalence varied by region, and was as high as 50–57% in some parts of Southern Nigeria [15]. The perceptions and use of TBA services differ from one socioeconomic, geographic, and cultural context to another. Hence, this study assessed the perceptions and factors associated with the use of TBA services in a rural local government area (LGA) in Lagos state, to shed more light on the rationale behind the continued patronage of TBA by pregnant women in Nigeria.

## Materials and methods

### Background information on the study area

Lagos State is one of the southwestern states in Nigeria with 20 Local Government Areas [16], of which 16 are urban and 4 rural. Ikorodu is one of the rural local government areas in the

state with a current population of 1,093,308 in 2023 of which 49.1% were males and 50.9% were females [17]. There were 59 registered TBAs in Ikorodu at the time the data was collected. Ikorodu LGA has a total number of 201 health facilities, including 145 primary and 56 secondary healthcare facilities.

## Study population, design, sample size determination, and selection of participants

This was a descriptive cross-sectional study conducted among pregnant women attending TBAs in Ikorodu LGA and who had resided in Ikorodu LGA for at least six months. Respondents were recruited between June 2019 and March 2020. Pregnant women who were ill during data collection were excluded from the study. The minimum sample size calculated was 295 using Cochran's formula [18], n = $Z^2pq/d2$ using p of 25.5% (number of pregnant women who utilized TBAs in rural areas 2018 NDHS) [19], and d of 0.05. The sample size was however increased by 10% to compensate for improperly completed questionnaires or opt-outs by any of the selected respondents. A total of 347 consented respondents were finally recruited for the study. From the list of the fifty-two registered TBAs in Ikorodu LGA, eighteen were selected by simple random sampling via balloting. To obtain the number of respondents for each traditional maternity clinic, the formula N/n was used, where N is the required sample size, and n is the number of TBAs, hence an average of nineteen (19) pregnant women were consecutively recruited from each selected TBA.

## Study instrument and data collection

Data were collected using a pretested and interviewer-administered semi-structured questionnaire which was adapted from similar studies [5, 11, 20–23] The questionnaire obtained information on respondents' socio-demographic characteristics, perception towards the use of TBAs, and reasons for the use of TBAs.

## Data analyses

Data were cleaned and analyzed with Statistical Product and Service Solutions (IBM-SPSS) version 25. Descriptive and inferential analyses were conducted with statistical significance p set at $<0.05$ (95% confidence level). The basis for the inclusion of variables into the logistic regression model after bivariate analysis was because bivariate analysis usually does not factor in how a variable could influence the other, and therefore cannot give an explanation for the relationship between the two variables, but only provide a description. Logistic regression uses relationships found in bivariate analysis to predict the value of one of those factors based on the other, usually with a finite outcome. A cut-off p-value of $<0.2$ was used for the inclusion of variables into the regression model after bivariate analysis [24].

Fifteen statements on a five-point Likert scale were used to assess the perception of respondents towards the use of TBA services. A score of 1 was assigned to strongly agree, 2- agree, 3-neither, 4- disagree, and 5- strongly disagree. The sum scores for each respondent were calculated and converted to percentages. Respondents with scores less than 70% were considered to have a positive perception (more supportive) and those with scores of 70% and above were considered to have a negative perception (less supportive) of TBA services. The higher the score, the more negative (less supportive) the perception of the respondents about TBA services. Positive perception of TBA services corresponds with a score of $<70\%$. Utilization of services was dichotomized into the use of TBA only or the use of a combination of TBA and healthcare facilities for ANC and delivery services.

## Ethical considerations

The Health Research Ethics Committee of the College of Medicine, University of Lagos, approved the study (ADM/DCST/HREC/APP/585). Written informed consent was obtained from the respondents before administering the questionnaires and they were assured of their right to refuse participation without any consequences. The respondents were assured of the anonymity of their participation and no name or any identifier was collected. The authors had no access to information that could identify individual participants during or after data collection.

## Results

### Socio-demographic characteristics of respondents and spouses

Table 1 shows the mean age of respondents was 29.2±5.7 years, with almost half 172(49.6%) between the ages of 26–34 years. The majority 337 (97.1%) of the respondents were married, about half 180 (51.9%), had secondary education, 179(51.6%) were manual skilled workers, and 226 (65.1%) earned less than ₦18,500 monthly. The mean age of the respondent's spouse/ partner was 37.1±8.7 years, and more than half 200 (59.3%) had secondary education.

**Perception of respondents towards the services of traditional birth attendants.** Table 2 shows that 99.2% of the respondents strongly agreed and agreed that TBAs' services achieved the expected results, and 93.4% of respondents also strongly agreed and agreed that TBAs' had the necessary skills to perform deliveries. Less than half of the respondents, 43.6% strongly agreed and agreed that TBAs use sterile equipment for delivery, and 78.7% strongly disagreed and disagreed that the care provided by the TBA cannot result in complications. Most 91.6% strongly agreed and agreed that TBAs should not be stopped and offered better privacy than hospitals 68.6%, provided better emotional and psychological support than hospitals 79.2%, birth defects can be detected by TBAs 75.8% and herbs provided by TBAs were more effective than orthodox drugs 64.2%. TBA facilities being standard were strongly agreed and agreed by 77.3% of the respondents, while more than half were neutral 56.5% to the statement that TBA homes were cleaner than hospitals. Most 68.6% strongly agreed and agreed that TBAs offered better privacy than hospitals and provided better emotional and psychological support than hospitals 79.2%.

Fig 1 shows that all (100%) of the respondents had a positive perception of the traditional birth attendant services.

Fig 2 shows that 283(81.6%) of the respondents utilized TBA services due to numerous positive recommendations from other users, 268(77.2%) used TBAs because of their welcoming and hospitable staff, while 261 (75.2%) attributed their use to the perceived spiritual protection offered, and 238(68.6%) used TBAs because they had used them in the past. Also, more than a quarter, 99(28.5%) attributed their use of TBA to their dislike for orthodox drugs and injections. Most of the respondents, 212(61.1%) preferred TBAs out of their fear of caesarian section, 190(54.8%) used TBAs because of the proximity of TBAs to their homes, and 157(45.2%) used TBAs because TBAs were cheaper than hospitals. Only 85(24.5%), 69(19.9%) and 54 (15.6%) respondents used TBAs due to the long waiting times at the health facility, difficult transportation to the hospital, and insufficient hospital staff respectively.

**Utilization of TBA services only in the current pregnancy.** Table 3 shows that 244 (70.3%) of the respondents utilized TBA services only in the current pregnancy. Other healthcare facilities utilized by those who combined TBA and healthcare facilities include private hospitals 37 (35.9%), general hospitals 32 (31.1%), PHC 30(29.1%), and faith-based clinics 4 (3.9%). More than half 195 (56.2%) of the respondents earned between ₦18,500-₦50,000

**Table 1. Socio-demographic characteristics of respondents and spouses.**

| Variable | Frequency (n = 347) | Percent (%) |
|---|---|---|
| **Age (years)** | | |
| 17–25 | 104 | 30.0 |
| 26–34 | 172 | 49.6 |
| 35–43 | 71 | 20.5 |
| Mean ± SD = 29.2±5.7 years | | |
| **Marital status** | | |
| Married/Cohabiting | 337 | 97.1 |
| Separated/Divorced | 4 | 1.2 |
| Single | 4 | 1.2 |
| Widowed | 2 | 0.6 |
| **Respondent's level of education** | | |
| No formal education | 24 | 6.9 |
| Primary | 94 | 27.1 |
| Secondary | 180 | 51.9 |
| Tertiary | 49 | 14.1 |
| **Respondent's employment status** | | |
| Professional | 25 | 7.3 |
| Manual Skilled | 179 | 51.6 |
| Intermediate skilled | 5 | 1.4 |
| Unskilled | 111 | 32 |
| Unemployed | 27 | 7.8 |
| **Respondents' average monthly income (₦)** | | |
| <18,500 | 226 | 65.1 |
| 18,500–50,000 | 116 | 33.4 |
| >50,000–100,000 | 5 | 1.4 |
| **Age of respondents' spouse (years) n = 337** | | |
| 21–36 | 161 | 47.8 |
| 37–52 | 151 | 44.8 |
| 53–68 | 25 | 7.4 |
| Mean ± SD | = 37.1±8.7 years | |
| **Educational qualification of respondents' spouse** (n = 337) | | |
| No formal education | 5 | 1.5 |
| Primary | 42 | 12.5 |
| Secondary | 200 | 59.3 |
| Tertiary | 90 | 26.7 |

household income monthly and spent 305 (87.9%) which is less than 10% of the income on TBA services. Only 60 (17.3%) had health insurance coverage. Most 144 (41.5%) respondents lived 1–2 km away from the TBA and 3–4km from the nearest health facility 126 (36.3%). It took less than 30 minutes for 184 (53.0%) to get to the TBA and 164 (47.3%) to the nearest health facility. TBA services were culturally acceptable by almost all 343 (98.9%) of the respondents.

**Factors influencing utilization of TBA services only in the current pregnancy.** Bivariate analysis showed that the highest proportion of respondents who utilized TBA services in the current pregnancy, 80.9% had primary education while the least, 38.8% had tertiary education and the difference in proportions was statistically significant, ($\chi^2$ = 31.896, p<0.001) While,

**Table 2. Perception of respondents towards the services of TBA\*.**

| Statement | Strongly Agreed (%) | Agreed (%) | Neither (%) | Disagreed (%) | Strongly Disagreed (%) |
|---|---|---|---|---|---|
| The services provided by the TBA achieve the expected results | 146 (42.1) | 198 (57.1) | 2 (0.6) | 1 (0.3) | 0 (0.0) |
| TBAs have the necessary skills to perform deliveries | 148 (42.7) | 176 (50.7) | 23 (6.6) | 0 (0.0) | 0 (0.0) |
| TBAs use sterile equipment for delivery | 21 (6.1) | 130 (37.5) | 106 (30.1) | 79 (22.8) | 11 (3.2) |
| TBAs can perform caesarian section | 1 (0.3) | 7 (2.0) | 62 (17.9) | 143 (41.2) | 134 (38.6) |
| The care provided by the TBA cannot result in complications. | 8 (2.3) | 28 (8.1) | 38 (11.0) | 197 (56.8) | 76 (21.9) |
| TBAs should be integrated into modern health care. | 100 (28.8) | 137 (39.5) | 48 (13.8) | 28 (8.1) | 34 (9.8) |
| TBAs should not be stopped | 168 (48.4) | 150 (43.2) | 29 (8.4) | 0 (0.0) | 0 (0.0) |
| TBAs can detect birth defects | 58 (16.7) | 205 (59.1) | 63 (18.2) | 19 (5.5) | 2 (0.6) |
| TBA facilities are standard | 44 (12.7) | 224 (64.6) | 78 (22.5) | 1 (0.3) | 0 (0.0) |
| TBA homes are cleaner than hospitals | 16 (4.6) | 82 (23.6) | 196 (56.5) | 47 (13.5) | 6 (1.7) |
| I trust the services of the TBA more than hospitals | 74 (21.3) | 157 (45.2) | 94 (27.1) | 20 (5.8) | 2 (0.6) |
| The herbs provided by the TBA are more effective than modern medicine. | 81 (23.3) | 142 (40.9) | 97 (28.0) | 25 (7.2) | 2 (0.6) |
| TBAs are older so they are more experienced than healthcare providers | 6 (1.7) | 8 (2.3) | 97 (28.0) | 156 (45.0) | 80 (23.1) |
| TBAs offer more privacy than hospitals | 48 (13.8) | 190 (54.8) | 85 (24.5) | 24 (6.9) | 0 (0.0) |
| TBAs provide better emotional and psychological support than hospitals. | 67 (19.3) | 208 (59.9) | 62 (17.9) | 9 (2.6) | 1 (0.3) |

\*A traditional birth attendant (TBA) is a person who assists the mother during childbirth and who initially acquired her skills by delivering babies herself or by working with other TBA.

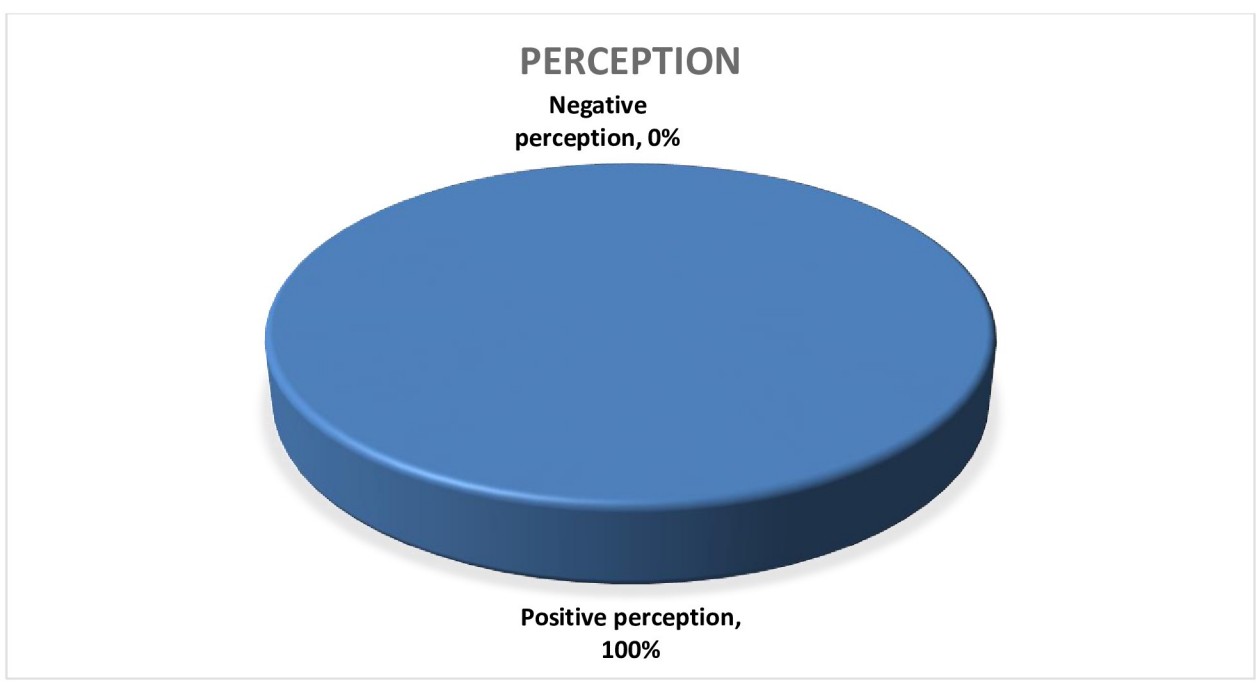

**Fig 1. Overall Perception of respondents towards the services of TBA.**

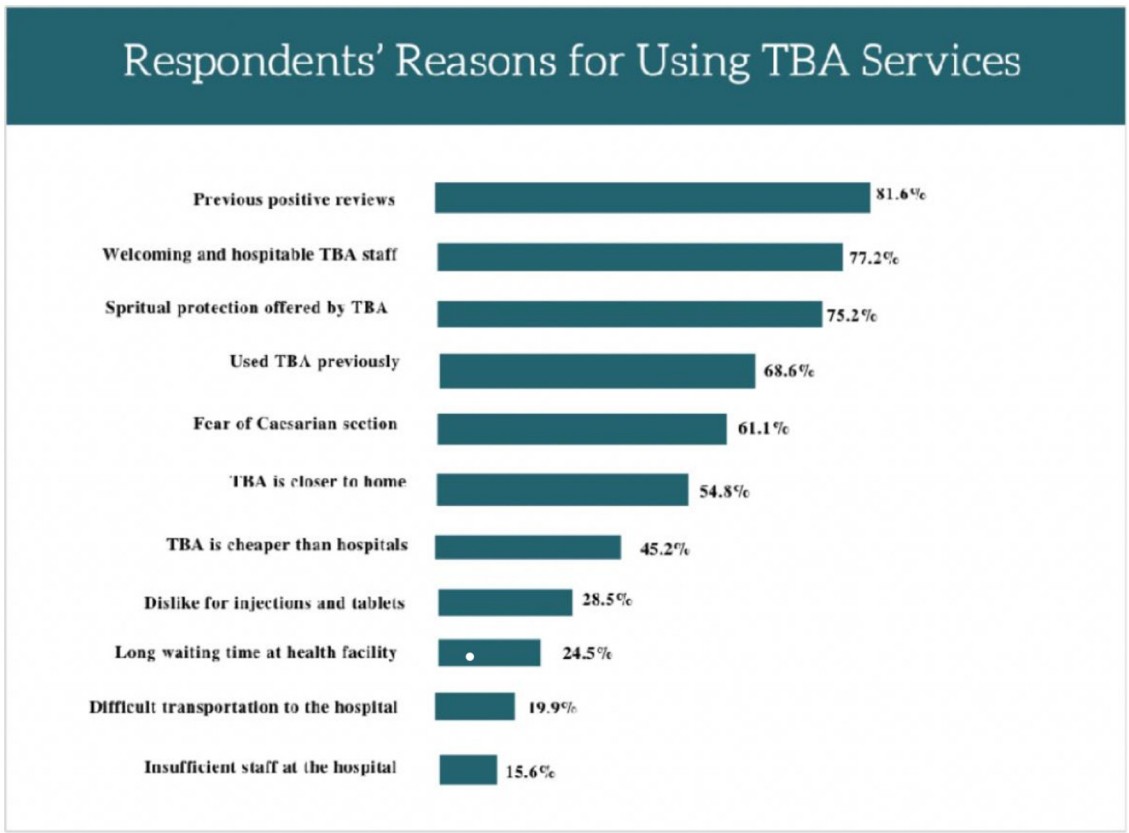

*Multiple responses

**Fig 2. Respondents' reasons for utilizing TBA services in the current pregnancy.**

the highest proportion of respondents who utilized only TBA services in the current pregnancy, 76.1% earned < ₦18,500 while the least, 40.0% earned >₦50,000–100,000 and the difference in proportions was statistically significant, ($\chi^2$ = 11.357, p = 0.003). Similarly, the highest proportion of respondents' spousal educational level who utilized only TBA services in the current pregnancy, 78.7% had no formal/primary education while the least, 48.9% had tertiary education and the difference in proportions was statistically significant, ($\chi^2$ = 26.175, p<0.001). Also, the higher proportion of the monthly household income of respondents who utilized only TBA services, 87.7% earned < ₦18,500 while the least, 53.3% earned >₦50,000–100,000 and the difference in proportions was statistically significant, ($\chi^2$ = 20.239, p<0.001). The highest proportion of respondents who utilized only TBA services, 76.3% had no insurance coverage while 41.7% had health insurance coverage and the difference in proportions was statistically significant, ($\chi^2$ = 28.529, p<0.001) (Table 4).

**Predictors of utilization of TBA services only in the current pregnancy.** The results indicate that respondents with higher levels of education were significantly less likely to utilize TBAs compared to those with no formal education. Those with secondary education had 73.9% lower odds of utilizing TBAs (aOR = 0.261; 95%CI; 0.108–0.629; p = 0.003) and those with tertiary education had 64.8% lower odds of TBA utilization (aOR = 0.352; 95%CI; 0.162–0.769; p = 0.009). Similarly, respondents without health insurance coverage had significantly

**Table 3. Utilization of TBA services only in current pregnancy among respondents.**

| Variable | Frequency (n = 347) | Percent (%) |
|---|---|---|
| **Health provider utilized by respondents in the current pregnancy** | | |
| TBA only | 244 | 70.3 |
| TBA and other health facilities | 103 | 29.7 |
| **Other health providers utilized by respondents (n = 103)** | | |
| Private hospital | 37 | 35.9 |
| General Hospital | 32 | 31.1 |
| Primary health care (PHC) | 30 | 29.1 |
| Faith-based clinic | 4 | 3.9 |
| **Monthly household income (₦)** | | |
| <18,500 | 65 | 18.7 |
| 18,500–50,000 | 195 | 56.2 |
| >50,000–100,000 | 77 | 22.2 |
| >100,000 | 10 | 2.9 |
| **Cost of utilizing TBA services** | | |
| <10% of household income | 305 | 87.9 |
| 10–20% of household income | 23 | 6.6 |
| >20% of household income | 19 | 5.45 |
| **Health insurance coverage** | | |
| Yes | 60 | 17.3 |
| No | 287 | 82.7 |
| **Estimated distance of TBA facility from respondents' home** | | |
| <1km | 40 | 11.5 |
| 1–2km | 144 | 41.5 |
| 3–4km | 125 | 36.0 |
| >5km | 38 | 11.0 |
| **Estimated distance of nearest health facility from respondents' home** | | |
| <1km | 67 | 19.3 |
| 1–2km | 96 | 27.7 |
| 3–4km | 126 | 36.3 |
| >5km | 58 | 16.7 |
| **Time taken by respondents to get to TBA** | | |
| <30 minutes | 184 | 53.0 |
| 30 minutes-1 hour | 125 | 36.0 |
| More than 1 hour | 38 | 11.0 |
| **Time taken to get to the health facility** | | |
| <30 minutes | 164 | 47.3 |
| 30 minutes- 1 hour | 125 | 36.0 |
| More than 1 hour | 58 | 16.7 |
| **Cultural acceptability of TBA** | | |
| Yes | 343 | 98.9 |
| No | 4 | 1.1 |

higher odds of utilizing TBAs compared to those with insurance coverage (aOR = 3.017; 95% CI; 1.476–6.166; p = 0.002). There were no statistically significant association found between respondents' monthly income, monthly household income, time taken to get to TBA, and the use of TBA services in the current pregnancy (Table 5).

**Table 4. Factors influencing utilization of TBA services only in current pregnancy.**

| Variables | TBA only (n = 244, 70.3%) | TBA and HF (n = 103, 29.7%) | Total (n = 347, 100%) | Test statistics |
|---|---|---|---|---|
| **Respondents' age** | | | | |
| 17–25 | 80 (76.9%) | 24 (23.1%) | 104 (100.0%) | $x^2$ = 3.217 |
| 26–34 | 115 (66.9%) | 57 (33.1%) | 172 (100.0%) | p = 0.200 |
| 35–43 | 49 (69.0%) | 22 (31.0%) | 71 (100.0%) | |
| **Respondents' level of education** | | | | |
| No formal education | 14 (58.3%) | 10 (41.7%) | 24 (100.0%) | $x^2$ = 31.896 |
| Primary | 76 (80.9%) | 18 (19.1%) | 94 (100.0%) | **p <0.001** |
| Secondary | 135 (75.0%) | 45 (25.0%) | 180 (100.0%) | |
| Tertiary | 19 (38.8%) | 30 (61.2%) | 49 (100.0%) | |
| **Respondent's employment status** | | | | |
| Professional | 14 (56.0%) | 11 (44.0%) | 25 (100.0%) | $x^2$ = 3.866 |
| Manual Skilled | 125 (69.8%) | 54 (30.2%) | 179 (100.0%) | P = 0.276 |
| Intermediate/ Unskilled | 87 (75.0%) | 29 (25.0%) | 116 (100.0%) | |
| Unemployed | 18 (66.7%) | 9 (33.3%) | 27 (100.0%) | |
| **Respondents' monthly income (₦)** | | | | |
| <18,500 | 172 (76.1%) | 54 (23.9%) | 226 (100.0%) | $x^2$ = 11.357 |
| 18,500–50,000 | 70 (60.3%) | 46 (39.7%) | 116 (100.0%) | p = **0.003** |
| >50,000–100,000 | 2 (40.0%) | 3 (60.0%) | 5 (100.0%) | |
| **Age of spouse** | | | | |
| 21–36 | 132 (72.5%) | 50 (27.5%) | 182 (100.0%) | $x^2$ = 1.178 |
| 37–52 | 92 (67.2%) | 45 (32.8%) | 137 (100.0%) | P = 0.551 |
| 53–68 | 12 (66.7%) | 6 (33.3%) | 18 (100.0%) | |
| **Spousal's level of education** | | | | |
| No formal /Primary education | 37 (78.7%) | 10 (21.3%) | 47 (100.0%) | $x^2$ = 26.175 |
| Secondary | 155 (77.5%) | 45 (22.5%) | 200(100.0%) | **p <0.001** |
| Tertiary | 44 (48.9%) | 46 (51.1%) | 90 (100.0%) | |
| **Respondents' monthly household income (₦)** | | | | |
| <18,500 | 57 (87.7%) | 8 (12.3%) | 65 (100.0%) | $x^2$ = 20.239 |
| 18,500–50,000 | 139 (71.3%) | 56 (28.7%) | 195 (100.0%) | **p <0.001** |
| >50,000–100,000 | 41 (53.3%) | 36 (46.8%) | 77 (100.0%) | |
| >100,000 | 7 (70.0%) | 3 (30.0%) | 10 (100.0%) | |
| **Cost of utilizing TBA services** | | | | |
| <10% of household income | 215 (70.5%) | 90 (29.5%) | 305 (100.0%) | $x^2$ = 1.666 |
| 10–20% of household income | 14 (60.9%) | 9 (39.1%) | 23 (100.0%) | p = 0.435 |
| >20–40% of household income | 15 (78.9%) | 4 (21.1%) | 19 (100.0%) | |
| **Respondents' health insurance coverage** | | | | |
| Yes* | 25 (41.7%) | 35 (58.3%) | 60 (100.0%) | $x^2$ = 28.529 |
| No | 219 (76.3%) | 68 (23.7%) | 287 (100.0%) | **P <0.001** |
| **Estimated time taken by respondents to get to TBA facility** | | | | |
| <30 minutes | 137 (74.5%) | 47 (25.5%) | 184 (100.0%) | $x^2$ = 3.363 |
| 30 minutes-1 hour | 83 (66.4%) | 42 (33.6%) | 125 (100.0%) | p = 0.186 |
| More than 1 hour | 24 (63.2%) | 14 (36.8%) | 38 (100.0%) | |

Significant p-value <0.05

**Table 5. Predictors of utilization of TBA services only in the current pregnancy.**

| Variables | aOR | 95%C I | p-value |
|---|---|---|---|
| **Respondents' level of education** | | | |
| No formal education* | 1 | | |
| Primary | 0.815 | 0.255–2.606 | 0.730 |
| Secondary | 0,261 | 0.108–0.629 | **0.003** |
| Tertiary | 0.352 | 0.162–0.769 | **0.009** |
| **Respondents' monthly income (₦)** | | | |
| <18,500 | 1 | | |
| 18,500–50,000 | 0.567 | 0.051–6.324 | 0.644 |
| >50,000–100,000 | 0.686 | 0.065–7.252 | 0.754 |
| **Monthly household income (₦)** | | | |
| <18,500 | 1 | | |
| 18,500–50,000 | 1.713 | 0.206–14.224 | 0.618 |
| >50,000–100,000 | 3.796 | 0.541–26.615 | 0.180 |
| >100,000 | 5.479 | 0.801–37.487 | 0.083 |
| **Respondents health insurance coverage** | | | |
| Yes* | 1 | | |
| No | 3.017 | 1.476–6.166 | **0.002** |
| **Time taken to get to TBA facility** | | | |
| <30 minutes | 1 | | |
| 30 minutes-1 hour | 0.677 | 0.289–1.584 | 0.369 |
| More than 1 hour | 0.917 | 0.390–2.159 | 0.843 |

*(reference group)

## Discussion

Despite the introduction of modern health facilities, safe motherhood initiative programs, free antenatal care services, and other maternal health services, evidence suggests that a significant number of deliveries are supervised by traditional birth attendants, especially in rural areas [5, 8, 22, 25]. Various studies have shown that the utilization of these traditional birth attendants has largely contributed to the burden of maternal and neonatal mortality in Africa [3, 12, 24–26]. The respondents in this study mainly were distributed between the ages of 26–34 years (49.6%), similar to a study conducted in Ogun state, Nigeria where 48.8% of respondents were between the ages of 26–35 years [11]. Also, a study conducted in South Sudan reported that 55.8% of the respondents were between 20–34 years [22]. However, this finding differs from the result of a study conducted in Kano state, Nigeria where 57.2% of the respondents were aged 19–29 [27]. This disparity can be explained by the fact that women in the Northern regions of the country get married earlier than their Western counterparts.

Almost all respondents (97.1%) were married or cohabiting, which is consistent with the findings of a similar study in Ogbomoso, Nigeria where 94.1% of the respondents were married [5], and the study in Ghana where 82.7% of the respondents were married [21] as well as a study in Ogun where 76.8% of the respondents were married [11]. This study revealed the significance of education in health-seeking behavior as about half (51.9%) of the respondents had primary/secondary education, and only a few 14.1% had tertiary education. This is similar to the findings of the study conducted in Ogun state, Nigeria where 54.1% of respondents had secondary education, 21.2% had primary education, and 20.0% had tertiary education [11].

This is however different from the findings of the study conducted in Bondo District Ghana, where 5.5% of respondents had secondary education, 32.7% had primary education, and 0.6% had tertiary education [21]. The findings in this study are not surprising since it was carried out in a rural local government area where women may have restricted access to higher education. The majority of the respondents earned less than ₦18,500 (65.1%), which is consistent with the findings of a similar study conducted in Ibadan, Nigeria, where 61.5% of respondents earned between ₦5,000 and ₦19,000 monthly [8]. This low income earned by women usually puts them in economically disadvantaged situations, especially in rural areas, and the cost of antenatal and delivery care can discourage the use of formal health care, especially if there are other 'cheaper' substitutes such as TBAs.

Perceptions are determined by an individual's beliefs about the potential outcomes of specific behaviour; if a negative outcome is expected, then the individual will have a negative perception and, if the outcome is beneficial, then such an individual will have a positive perception of that behavior [11]. Almost all (99.2%) respondents strongly agreed and agreed that TBAs' services achieved the expected results. A majority (93.4%) strongly agreed and agreed that TBAs have the necessary skills to conduct deliveries. These findings are similar to the findings of the study conducted in Ogun state, Nigeria [11]. A little below half (43.6%) of respondents agreed and strongly agreed that TBAs use sterile equipment for delivery, while 26.0% of respondents disagreed with the same statement, and 30% were indifferent. This finding is higher than that of a similar study conducted in Ibadan, Nigeria where 19.2% of the respondents agreed that TBAs use sterile equipment for their service [8]. However, only 8 (2.3%) of respondents strongly agreed with the statement that TBAs can perform caesarian sections, while a majority (79.8%) disagreed and strongly disagreed with the statement. This is relatively similar to the findings of the study conducted in Ogun state where 5.6% of the respondents agreed that TBAs can perform gynaecological surgeries [11]. Also, most of the respondents (68.3%) believed that TBAs should be integrated into modern health care. Most respondents (78.7%) disagreed and strongly disagreed that the care provided by TBAs could result in complications, and 91.6% of respondents agreed and strongly agreed that TBAs should not be stopped. This finding is similar to that of the study in Ibadan where 66.9% of respondents opposed banning TBAs [8]. However, this is in contrast to the findings of another study conducted in Southern Nigeria, where 85.7% of respondents believed that TBAs should be stopped from offering services [28]. This difference might be because respondents in this study were current users of TBAs and were recruited directly from their homes, while the contrast study respondents were recruited from health facilities. Furthermore, the majority (77.3%) of respondents agreed and strongly agreed that TBA services were standard. This finding differs from the results of another study in Southern Nigeria where 77.9% of respondents believed that TBA services were not standard [28]. This may be due to the differences in the study respondents. Furthermore, 64.2% of respondents agreed and strongly agreed that the herbs provided by the TBA are more effective than modern medicine in the present study. The study in Ogun state also reported that 79.2% of respondents believed TBA services were more effective than orthodox care [11]. Overall, all of the respondents (100%) had an overall positive perception of traditional birth attendants, which is similar to the results of the Ogun state study where findings revealed a positive perception and use of TBA services by the respondents [11]. However, this differs from the results of the study in Southern Nigeria where 37 (8.8%) and 66(15.7%) had very good and good perceptions of TBA respectively [28]. The 100% positive perception of respondents in this study could be because they were current users of traditional birth homes, which is indicative of their acceptance of the use of TBAs. Although some pregnant women may be aware of the safety of utilizing orthodox health facilities, the

strong pull of traditions motivates them to use TBA services, while the pull factors of modern health care and its quality care seem insignificant [29].

As regards the reasons for the use of TBA by respondents in this study, more than half (54%) of the respondents chose to use TBA services only because of the fear of the caesarian section at healthcare facilities. This finding is in keeping with other similar studies conducted in Rural Bangladesh [30], Lao [31], Kenya [32], and North-west Nigeria [33], where participants of the focused group discussions attributed their use of TBA services to their fear of surgeries and invasive procedures conducted at hospitals. Close to half of the respondents (45.2%) used TBA because of its being cheaper, this finding is similar to that of the study conducted in Ogun State, Nigeria where 50.9% of participants utilized the services of TBAs because they were cheaper [11]. Similarly, the study in Oyo state, Nigeria reported that 68.8% of the respondents utilized TBA services because they were cheaper [8]. Another reason stated by the majority of the respondents (77.0%) in this study for utilizing TBA services includes their welcoming attitude and hospitality of the staff, which is similar to the results from a study in Oyo state, where 80.7% of respondents attributed their use of TBAs to the compassionate care provided by TBA [26]. This might be because the TBA usually resides within the community, and effectively communicates using the local language, sharing similar values and norms with the community. Thus, cultivating some level of familiarity between the TBA and pregnant women in the community. This is corroborated by the fact that almost all respondents in this study confirmed the cultural acceptability of TBA. They uphold the traditions and preserve the culture of communities [34]. It has been indicated that culture, compassionate attitude toward TBA, non-availability of health facilities, and distance from the health facility were associated with the use of TBA services [11].

The place of delivery is a determinant factor for maternal mortality and morbidity [15]. Most (70.3%) of the respondents in this study utilized TBA services only during the current pregnancy for ANC and delivery services while a little above a quarter (29.7%) utilized TBA along with health facilities services such as private hospitals, general hospitals, and primary health care centers. This finding is similar to that of a similar study in Ogbomoso, Oyo state, Nigeria where 68.5% of the respondents were full utilizers of TBA [5].

Respondents' level of education was significantly associated with the use of TBA services only in this study. Those with lower levels of education utilized TBA services only compared with those with post-secondary education. This is similar to the findings of studies conducted in Ogbomoso [5], and Kenya [32]. However, on the contrary, respondents' level of education was not found to be statistically significant with the use of TBA in the Ibadan study [8]. Also, a statistically significant association was found between the monthly income of respondents and the use of TBA services only, respondents who earned less than ₦18,500 were more likely to use TBA services only compared to those who earned more. This is however, in contrast to the findings from other studies conducted in Kenya [32], and Ibadan [8] where the mother's income level was not statistically significant to the use of TBA services. This variation may be due to the different settings of the studies. Furthermore, the educational qualification of the spouse was significantly associated with the use of TBA services, respondents whose spouse had secondary education used TBA services only more than those with post-secondary education. This is similar to the findings of a study in Ghana where the educational qualification of the spouse was statistically significant to the use of TBA services [21]. Also, monthly household income was statistically significant with utilization of TBA services, which is consistent with that of the studies conducted in Ogbomoso [5], and South Sudan [22]. Furthermore, possession of health insurance coverage was associated with the use of TBA services only in this study, respondents without health insurance coverage used TBA services only more compared to those who had insurance coverage. This finding corroborates that of studies conducted in

Ghana and Indonesia where it was found that pregnant women without insurance coverage were more likely to use TBA services [12, 21]. The major healthcare financing system in Nigeria is out-of-pocket due to poor insurance coverage, especially in the rural areas, and as a result of the absence of wide coverage health insurance in the country, financial constraints have been a major barrier to accessing quality maternal health services, so it is not unusual that many pregnant women might opt for cheaper alternatives like TBAs [25]. This study also found that a good number of the respondents earned below the minimum wage in Nigeria (₦18,500). Hence, the cost of antenatal and delivery care can discourage the use of formal health care, especially if there are other substitutes such as the TBAs.

Binary logistic regression analysis showed that respondents' level of education and lack of health insurance coverage were independent predictors of using TBA-only services in the current pregnancy. The results indicate that respondents with higher levels of education were significantly less likely to utilize TBAs compared with those who had no formal education. This finding supports that of a study that reported that higher parental education, maternal employment, belonging to wealthy households, and having money to pay for health services have been reported to be associated with lower odds of TBA use [15]. Other studies in Ghana [35], Kenya [36], and Afghanistan [37] also confirmed higher maternal education influences the use of skilled birth attendants. Furthermore, this study found that respondents without health insurance coverage in their current pregnancy had significantly higher odds of utilizing TBAs compared to those who did not. This finding is in support of a similar study which reported that apart from household wealth status, having financial savings was a strong predictor of skilled-birth-attendant-assisted delivery [38], because having access to financial savings may cover the considerable expenses associated with maternal health service utilization, including transportation, laboratory tests, medications, etc.

The cross-sectional nature of the study does not allow for causal inferences. Hence, causation cannot be ascribed to the relationships found in this study. Moreover, the study took place among pregnant women already utilizing TBA services in their current pregnancy. However, the study adds to the body of evidence on the perception of pregnant towards utilization of TBA services and findings may apply to other similar community studies.

## Conclusion

This study revealed a positive perception of TBA by all respondents. Some of the cited reasons for the utilization of TBA services include past use of the services, positive recommendations of TBA from previous users, welcoming and hospitable staff, fear of caesarian section, and spiritual protection. The majority of the respondents in this study used TBA services only during the current pregnancy. Predictors of using only TBA services among respondents were respondents' (lower) level of education and lack of health insurance coverage. Continuous training and implementation of effective monitoring systems of TBA services by the government and other stakeholders is recommended to improve women's birthing experiences, especially in rural communities.

## Supporting information

**S1 Appendix. Study data set.**
(XLS)

## Acknowledgments

The authors appreciated the time and commitment of the respondents involved in the study.

## Author Contributions

**Conceptualization:** Esther O. Oluwole.

**Data curation:** Esther O. Oluwole, Tolulope E. Oluwadumila.

**Formal analysis:** Esther O. Oluwole.

**Investigation:** Esther O. Oluwole, Ifeoma P. Okafor, Abiola O. Temitayo-Oboh.

**Methodology:** Esther O. Oluwole, Ifeoma P. Okafor, Abiola O. Temitayo-Oboh.

**Project administration:** Esther O. Oluwole, Tolulope E. Oluwadumila, Ifeoma P. Okafor, Abiola O. Temitayo-Oboh.

**Software:** Esther O. Oluwole.

**Supervision:** Esther O. Oluwole.

**Writing – original draft:** Esther O. Oluwole.

**Writing – review & editing:** Esther O. Oluwole, Tolulope E. Oluwadumila, Ifeoma P. Okafor, Abiola O. Temitayo-Oboh.

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
