## [Decision Letter · Decision Letter 0]

25 Jul 2023

PONE-D-23-19072Exploring the perception and patterns of utilization of traditional birth attendants by pregnant women in a rural local government area of Lagos, NigeriaPLOS ONE

Dear Dr. Oluwole,

Thank you for submitting your manuscript to PLOS ONE. After careful consideration, we feel that it has merit but does not fully meet PLOS ONE’s publication criteria as it currently stands. Therefore, we invite you to submit a revised version of the manuscript that addresses the points raised during the review process.

We look forward to receiving your revised manuscript.

Kind regards,

Adetayo Olorunlana, Ph.D.

Academic Editor

PLOS ONE

Journal Requirements:

Reviewers' comments:

Reviewer's Responses to Questions

**Comments to the Author**

1. Is the manuscript technically sound, and do the data support the conclusions?

Reviewer #1: Yes

Reviewer #2: Yes

2. Has the statistical analysis been performed appropriately and rigorously? 

Reviewer #1: Yes

Reviewer #2: No

3. Have the authors made all data underlying the findings in their manuscript fully available?

Reviewer #1: Yes

Reviewer #2: Yes

4. Is the manuscript presented in an intelligible fashion and written in standard English?

Reviewer #1: Yes

Reviewer #2: Yes

5. Review Comments to the Author

Reviewer #1: This study provides relevant information on the perception of traditional birth attendants among pregnant women in a rural local government area in Lagos state. It was found that all the women had positive attitude towards TBAs and recommended continuous training and effective monitoring of TBAs. This is an interesting study that reveals rural women’s perceptions of TBA and their reasons for utilising them. This study is very topical, and findings can contribute towards strategies aimed at reducing maternal mortality especially in rural areas.

Please take note of the comments below to improve the manuscript.

1. Line 61 - 65: “TBAs help with initiating breastfeeding; providing health education on sexually transmitted illnesses (STIs), reproductive health, and nutrition; visiting mothers during and shortly following delivery to check for and educate them on the associated danger signs; and accompanying referrals to the health facilities for complicated deliveries”

Please note that this are not statutory functions of TBAs. This could be rephrased as: In certain settings, TBAs have been trained to perform other services which include helping with initiating breastfeeding ………

2. Line 125 – 127: “Respondents with scores of 0- 69% were considered to have a positive perception, and those with scores of 70% and above were considered to have a negative perception of TBAs. The higher the score, the more negative the perception of the respondents about TBA”.

Since a Likert scoring system was used, the lowest score obtainable is 15. It is thus not correct to state that scores of scores of 0- 69% were considered to have a positive perception, Rather state that scores less than 70% were considered to have….

3. Results: In the reporting of the results, the figures are presented differently in text e.g. 337 (97.1%) vs (344, 99.2%). Please be consistent.

4. Tables: Monthly household income (₦). Please write Naira in full.

5. Results: This study presents very important information on TBA in the rural community. However, there is no sound justification in differentiating between women who used only TBA and those that used TBA and health facility in the bivariate and multivariate analysis. Those that used both TBA and health facilities could have delivered their babies at the TBA. I will suggest focusing this study on the sociodemographic characteristics of women using TBAs, reasons for use of TBA and attitude of women that use TBA in Lagos. The perception of TBAs is already well elaborated in this paper, in the results and discussion, which is a strength of the paper.

6. Line 263. “That TBAs use sterile equipment for their service.8 However, only 8(2.3%) of respondents strongly agreed”

Please remove the number 8.

Reviewer #2: Review comments

General comments

The Authors are commended for conceptualizing this study and their intention to publish their research work. After going through the manuscript, I am of the opinion that there are certain aspects of the study that should be clarified

1. Concept/Title: The study was carried out among women attending ‘antenatal care’ from traditional birth attendants. The title of the manuscript did not bring this into focus. I wonder why assessing the perception of pregnant women on utilization of traditional birth attendants should be carried out among women already utilizing that service. I would have preferred such a concept to be a community based study. In line with the concept of the study, Authors should justify why this study is necessary.

2. Outcome variable: The outcome variable(s) of the study are not well defined. The rationale for using <70% of scores of respondents to categorize perception of respondents is not clear. Again, one wonders the relevance of pattern of utilization as determined in this study to current and future studies involving traditional birth attendants and to the body of knowledge generally. Authors should justify why that outcome variable is of interest.

3. Authors utilized data involving the spouses of the respondents who are less than the sample size of 347 in logistic regression analysis. This is wrong and should be reviewed. The entire results of the logistic regression analysis should be reviewed.

4. Authors should pay attention to guidelines of the Journal in presenting this manuscript especially the abstract section. Also, in presenting the logistic regression analysis in the abstract, Authors should use adjusted odd ratio and 95% confidence interval.

5. The background of the study is well written and Authors are commended for their great efforts.

The responses to the comments above will actually determine the merit of the manuscript in proceeding with the review process.

6. PLOS authors have the option to publish the peer review history of their article (what does this mean?). If published, this will include your full peer review and any attached files.

Reviewer #1: No

Reviewer #2: **Yes: **EDMUND NDUDI OSSAI

---

## [Author Response · Author response to Decision Letter 0]

28 Aug 2023

Reviewers’ comments Authors’ comments Page Line Number

Reviewer #1: Review comments

Line 61 - 65: “TBAs help with initiating breastfeeding; providing health education on sexually transmitted illnesses (STIs), reproductive health, and nutrition; visiting mothers during and shortly following delivery to check for and educate them on the associated danger signs; and accompanying referrals to the health facilities for complicated deliveries”

Please note that this are not statutory functions of TBAs. This could be rephrased as: In certain settings, TBAs have been trained to perform other services which include helping with initiating breastfeeding ………

 Correction has been effected 3 Line 60 - 61

Line 125 – 127: “Respondents with scores of 0- 69% were considered to have a positive perception, and those with scores of 70% and above were considered to have a negative perception of TBAs. The higher the score, the more negative the perception of the respondents about TBA”.

Since a Likert scoring system was used, the lowest score obtainable is 15. It is thus not correct to state that scores of scores of 0- 69% were considered to have a positive perception, Rather state that scores less than 70% were considered to have….

 Correction has been effected 6 Line 120 – 121

Results: In the reporting of the results, the figures are presented differently in text e.g. 337 (97.1%) vs (344, 99.2%). Please be consistent.

 Correction has been effected 7 Line 136-137

Tables: Monthly household income (₦). Please write Naira in full.

 Correction has been effected 7 Line 140

Results: This study presents very important information on TBA in the rural community. However, there is no sound justification in differentiating between women who used only TBA and those that used TBA and health facility in the bivariate and multivariate analysis. Those that used both TBA and health facilities could have delivered their babies at the TBA. I will suggest focusing this study on the sociodemographic characteristics of women using TBAs, reasons for use of TBA and attitude of women that use TBA in Lagos. The perception of TBAs is already well elaborated in this paper, in the results and discussion, which is a strength of the paper. The study title has been modified as 

“Exploring the perception, utilization, and reasons for use of traditional birth attendants among rural dwelling pregnant women in Lagos, Nigeria”

 1 Line 1-2

Line 263. “That TBAs use sterile equipment for their service.8 However, only 8(2.3%) of respondents strongly agreed”

Please remove the number 8. Corrected. The “8” refers reference No 8. Line 263.

Reviewer #2: Review comments

Concept/Title: The study was carried out among women attending ‘antenatal care’ from traditional birth attendants. The title of the manuscript did not bring this into focus.

 I wonder why assessing the perception of pregnant women on utilization of traditional birth attendants should be carried out among women already utilizing that service. I would have preferred such a concept to be a community based study. In line with the concept of the study, Authors should justify why this study is necessary The title has been modified to bring this into focus as; 

“Exploring the perception, utilization, and reasons for use of traditional birth attendants among rural dwelling pregnant women in Lagos, Nigeria.”

Justification

Over half of women and men in Nigeria live in rural areas (54%), and 56% of rural women were more likely to have received ANC from a skilled provider and 29% and 26% of births to rural mothers were assisted by a skilled provider and were delivered in

a health facility respectively. (NDHS 2018)

Several similar studies have been carried out in communities but none among those utilizing TBAs. And we could only get our sample size using alternative source of care by going to the TBA centres, since our respondents were pregnant women, it might have been a bit difficult to recruit such no of pregnant women in communities, especially due to our culture and tradition.

This is why we decided to recruit pregnant women who were already utilizing TBA homes. 1 Line 1-2

Outcome variable: The outcome variable(s) of the study are not well defined. The rationale for using <70% of scores of respondents to categorize perception of respondents is not clear. 

Again, one wonders the relevance of pattern of utilization as determined in this study to current and future studies involving traditional birth attendants and to the body of knowledge generally. Authors should justify why that outcome variable is of interest.

 Justification

The outcome variable for this study were; perception, utilization (TBA only/TBA and HFs) and reasons for use of TBA. 

The rationale for using <70% was adapted and modified from studies “ https://doi.org/10.1016/j.jsps.2022.01.008

https://doi.org/10.1371/journal.pgph.0000171.t001

“The pattern of utilization” has been modified accordingly, as Utilization of health care services by respondents. 

Authors utilized data involving the spouses of the respondents who are less than the sample size of 347 in logistic regression analysis. This is wrong and should be reviewed. 

The entire results of the logistic regression analysis should be reviewed. Reasons: 

This is because only 337 respondents were married and had spouses at the time the study was conducted. 

4 respondents were separated/divorced; 

4 were single and

2 were widowed.

Comments.

The entire results of the logistic regression analysis has been be reviewed appropriately. 

Authors should pay attention to guidelines of the Journal in presenting this manuscript especially the abstract section. Also, in presenting the logistic regression analysis in the abstract, Authors should use adjusted odd ratio and 95% confidence interval.

Attention has now been paid to the journal guidelines in presenting this manuscript.

adjusted odd ratio and 95% confidence interval has been effected.

The background of the study is well written and Authors are commended for their great efforts.

 Thank you and this is well appreciated sir/ma.

---

## [Decision Letter · Decision Letter 1]

26 Oct 2023

PONE-D-23-19072R1Exploring the perception, utilization, and reasons for use of traditional birth attendants among rural dwelling pregnant women in Lagos, Nigeria.PLOS ONE

Dear Dr. Oluwole,

Thank you for submitting your manuscript to PLOS ONE. After careful consideration, we feel that it has merit but does not fully meet PLOS ONE’s publication criteria as it currently stands. Therefore, we invite you to submit a revised version of the manuscript that addresses the points raised during the review process.

Kindly pay attention to the methodology and the concept use for consistency.

We look forward to receiving your revised manuscript.

Kind regards,

Adetayo Olorunlana, Ph.D.

Academic Editor

PLOS ONE

Journal Requirements:

Reviewers' comments:

Reviewer's Responses to Questions

**Comments to the Author**

1. If the authors have adequately addressed your comments raised in a previous round of review and you feel that this manuscript is now acceptable for publication, you may indicate that here to bypass the “Comments to the Author” section, enter your conflict of interest statement in the “Confidential to Editor” section, and submit your "Accept" recommendation.

Reviewer #1: All comments have been addressed

Reviewer #2: (No Response)

2. Is the manuscript technically sound, and do the data support the conclusions?

Reviewer #1: Yes

Reviewer #2: Yes

3. Has the statistical analysis been performed appropriately and rigorously? 

Reviewer #1: Yes

Reviewer #2: No

4. Have the authors made all data underlying the findings in their manuscript fully available?

Reviewer #1: Yes

Reviewer #2: Yes

5. Is the manuscript presented in an intelligible fashion and written in standard English?

Reviewer #1: Yes

Reviewer #2: No

6. Review Comments to the Author

Reviewer #1: All comments have been addressed appropriately. The title is better phrased. The results section has been written with consistency.

Reviewer #2: Review comments

General comments

Authors are commended for the good review of the manuscript. However, further revision will also be required before the manuscript could be accepted for publication. Authors should take note f the following points.

Title

The title gives the impression that this was a community based study.

The word ‘Exploring’ to a large extent suits qualitative studies more.

Authors should also note previous comments in this regard.

Abstract

Majority (70.3%) received antenatal care and delivery at TBA only.

Based on the statement above, Have the women delivered during the period of study? The statement also counters the title of the study. These should be clarified.

It will be good to state the outcome measure of the study in the abstract

There may be no need to report the factors since binary logistic regression analysis was performed

Materials and methods

Study area

Use the current population projection for Ikorodu LGA.

Do we justify the two stage process for recruitment of respondents since consecutive recruitment was used in recruiting the respondents?

Data analysis

Use IBM-SPSS and SPSS should be written in full

Figure 2

Reasons for using TBA services. Were the responses based on multiple response option? This should be clarified.

In the abstract , include the first four reasons.

Table 4

Since p value is a probability, p values of 0.000 should be presented as <0.001

Housewife is not an example of employment status, instead use unemployed.

Table 5

Authors did not indicate the basis for inclusion of variables into the logistic regression model after bivariate analysis. This should be stated.

Variables related to the spouses are less than the sample size included for the study and should be removed from the logistic regression model.

What are the implications of the adjusted odds ratios reported as negative figures?

Authors should include Study limitations before the conclusion section. It should be emphasized that this study took place at traditional birth homes by individuals already utilizing their services during pregnancy.

Conclusion

The use of ‘Respondents’ choice of health services’ utilization’ in this section is confusing and should be deleted. Appropriate terms as previously used in the manuscript should be maintained.

7. PLOS authors have the option to publish the peer review history of their article (what does this mean?). If published, this will include your full peer review and any attached files.

Reviewer #1: No

Reviewer #2: **Yes: **EDMUND NDUDI OSSAI

---

## [Author Response · Author response to Decision Letter 1]

30 Oct 2023

Title

The title gives the impression that this was a community based study.

The word ‘Exploring’ to a large extent suits qualitative studies more.

Authors should also note previous comments in this regard.

 The title of the manuscript has been revised accordingly

“Perception and reasons for utilization of antenatal care services among rural pregnant women attending traditional birth attendants' homes in Lagos, Nigeria”

 1 Line 1 - 2

Abstract

Majority (70.3%) received antenatal care and delivery at TBA only.

Based on the statement above, Have the women delivered during the period of study? The statement also counters the title of the study. These should be clarified.

It will be good to state the outcome measure of the study in the abstract

There may be no need to report the factors since binary logistic regression analysis was performed

 …..and majority (70.3%) utilized only TBA while 29.7% combined both TBA and healthcare facilities for healthcare services in the current pregnancy. 

The statement is clarified.

The outcome measures of this study were perception, utilization and the reason for use f traditional birth attendants among rural dwelling pregnant women in Lagos, Nigeria. 

This has been stated.

 2 Line 37 – 39

Line 34-36

Materials and methods

Study area

Use the current population projection for Ikorodu LGA.

 a current population of 1,093,308 in 2023of which 49.1%were males and 50.9% were females.

This has been adjusted.

 5 Line 92-93

Do we justify the two stage process for recruitment of respondents since consecutive recruitment was used in recruiting the respondents?

Data analysis

Use IBM-SPSS and SPSS should be written in full Adjusted appropriately

From the list of the fifty-two registered TBA in Ikorodu LGA, eighteen were selected by simple random sampling via balloting. To obtain the number of respondents for each traditional maternity clinic, the formula N/n was used, where N is the required sample size, and n is the number of TBAs, hence an average of nineteen (19) pregnant women were consecutively recruited from each of the selected TBA over three weeks.

International Business Machines Corporation: Statistical Package For The Social Sciences (IBM-SPSS)

 5

6 Line 105-106

Line 117-118

Figure 2

Reasons for using TBA services. Were the responses based on multiple response option? This should be clarified.

In the abstract , include the first four reasons.

 Yes. It was based on multiple responses. This is indicated on the figure now.

Done. Recommendations from previous users (81.6%), welcoming and hospitable staff (77.2%), spiritual protection (75.2%) and past use (68.6%), were some of the reasons cited by the respondents for the use of TBA services. 9

1 Figure 2

Line 40-42

Table 4

Since p value is a probability, p values of 0.000 should be presented as <0.001

Housewife is not an example of employment status, instead use unemployed Done

Done 13 

Table 5

Authors did not indicate the basis for inclusion of variables into the logistic regression model after bivariate analysis. This should be stated.

Variables related to the spouses are less than the sample size included for the study and should be removed from the logistic regression model.

What are the implications of the adjusted odds ratios reported as negative figures?

 Bivariate analysis usually does not factor in how a variable could influence the other, and therefore cannot give an explanation for the relationship between the two variables, but only provide a description. Logistic regression uses relationship found in bivariate analysis to predict the value of one of those factors based on the other, usually with a finite outcomes.

Negative odd ratios indicate the event is less likely to occur. 14 Line 219-220

Authors should include Study limitations before the conclusion section. It should be emphasized that this study took place at traditional birth homes by individuals already utilizing their services during pregnancy.

 Included:

“Causation cannot be ascribed to the relationships found in this study. Moreover, the study took place among pregnant women already utilizing TBA services.”

 20 Line 348-349

Conclusion

The use of ‘Respondents’ choice of health services’ utilization’ in this section is confusing and should be deleted. Appropriate terms as previously used in the manuscript should be maintained Corrected:

Predictors of healthcare services utilization among respondents were……………… 20 Line 357-358

---

## [Decision Letter · Decision Letter 2]

20 Nov 2023

PONE-D-23-19072R2Perception and reasons for utilization of antenatal care services among rural pregnant women attending traditional birth attendants' homes in Lagos, NigeriaPLOS ONE

Dear Dr. Oluwole,

Thank you for submitting your manuscript to PLOS ONE. After careful consideration, we feel that it has merit but does not fully meet PLOS ONE’s publication criteria as it currently stands. Therefore, we invite you to submit a revised version of the manuscript that addresses the points raised during the review process.

We look forward to receiving your revised manuscript.

Kind regards,

Adetayo Olorunlana, Ph.D.

Academic Editor

PLOS ONE

Journal Requirements:

Reviewers' comments:

Reviewer's Responses to Questions

**Comments to the Author**

1. If the authors have adequately addressed your comments raised in a previous round of review and you feel that this manuscript is now acceptable for publication, you may indicate that here to bypass the “Comments to the Author” section, enter your conflict of interest statement in the “Confidential to Editor” section, and submit your "Accept" recommendation.

Reviewer #2: (No Response)

2. Is the manuscript technically sound, and do the data support the conclusions?

Reviewer #2: Yes

3. Has the statistical analysis been performed appropriately and rigorously? 

Reviewer #2: Yes

4. Have the authors made all data underlying the findings in their manuscript fully available?

Reviewer #2: Yes

5. Is the manuscript presented in an intelligible fashion and written in standard English?

Reviewer #2: Yes

6. Review Comments to the Author

Reviewer #2: Review comments

General comments

I commend the efforts of the Authors in reviewing this manuscript. I am also of the opinion that certain aspects of the manuscript need to undergo further revision.

Authors should take note of the following observations.

1. Title of study: I suggest that the Authors should consider the suggestion below

Perception and reasons for choice of informal provider among women receiving antenatal care from traditional birth attendants in rural communities of Lagos state, Nigeria

2. Abstract

The acronym TBA was not introduced in the abstract.

Define the outcome variable, ‘utilization’ in the methods section so the result of regression analysis will be meaningful.

Include also in the methods section, how positive perception was determined.

Use; Predictors of use of TBA only included:

3. Methods

Study population

Line 94; …….., attending TBAs. This should be reviewed.

Line 99, p of 25.5% is a proportion and not a number

4. Data analysis

Use the new name for SPSS, Statistical Product and Service Solutions

What was the rationale for using scores <70% as having a positive perception. Any justification? Any reference?

Authors should also take another look at the scoring system for the variables. For example, TBAs can perform Caesarian section.

Utilization as an outcome variable was not explained in the Methods section.

Authors should use variables with p values <0.2 at bivariate analysis as basis for inclusion of variables into the binary logistic regression model.

Refer to, Maldonado G, Greenland S. Simulation study of confounder-selection strategies. Am J Epidemiol. 1993;138:923-36.

5. Results

Socio-demographic characteristics

Line 135, The proportion 51.9% cannot be regarded as ‘about half’.

Table 1

Use Percent instead of Percentage

Include (n=317) for variables related to the spouses

6. Discussion

Page 317, should be use of TBA only

Page 318; Those with secondary education and lower used TBA only compared with those with post secondary education. This statement should be reviewed.

There is no need to discuss results of bivariate analysis when that of multivariate analysis are included in the same study.

Line 352; This study was community based with…….. This statement is not correct. The study is like a ‘health facility based’ study and not a community based study. The study took place among those assessing care among traditional birth attendants.

7. Limitation

Delete the first statement under this sub-heading. Keep the second statement. Recall the 100% positive perception of the respondents as obtained in this study and state that it may be due to the study being conducted among those assessing care among TBAs.

8. References

Authors should ensure that referencing style conforms to Journal guidelines.

7. PLOS authors have the option to publish the peer review history of their article (what does this mean?). If published, this will include your full peer review and any attached files.

Reviewer #2: **Yes: **EDMUND NDUDI OSSAI

---

## [Author Response · Author response to Decision Letter 2]

3 Jan 2024

Response to Reviewers

PONE-D-23-19072R1 05-12-2023

Perception and reasons for choice of informal provider among women receiving antenatal care from traditional birth attendants in rural communities of Lagos state, Nigeria

Reviewer #2: Review comments

Reviewers’ comments Authors’ comments Page Line Number

Title

I suggest that the Authors should consider the suggestion below

Perception and reasons for choice of informal provider among women receiving antenatal care from traditional birth attendants in rural communities of Lagos state, Nigeria

 The title of the manuscript has been revised accordingly

“Perception and reasons for choice of informal provider among women receiving antenatal care from traditional birth attendants in rural communities of Lagos state, Nigeria”

 1 Line 1 - 2

Abstract

The acronym TBA was not introduced in the abstract.

Define the outcome variable, ‘utilization’ in the methods section so the result of regression analysis will be meaningful.

Include also in the methods section, how positive perception was determined.

Use; Predictors of use of TBA only included: 

 This has been introduced

Done

Done 2

5-6

6 Line 28 – 29

Line 115 – 120

Line 28 – 29

Methods

Study population

Line 94; …….., attending TBAs. This should be reviewed.

Line 99, p of 25.5% is a proportion and not a number

This has been adjusted accordingly

 5

5 Line 94-95

Line 98

Data analysis

Use the new name for SPSS, Statistical Product and Service Solutions

What was the rationale for using scores <70% as having a positive perception. Any justification? Any reference?

Authors should also take another look at the scoring system for the variables. For example, TBAs can perform Caesarian section. 

Utilization as an outcome variable was not explained in the Methods section.

Adjusted appropriately “Statistical Product and Service Solutions”

The choice of using scores below 70%, to interpret Likert scale responses as having a positive perception is a subjective decision made by the researchers based on our judgment of the specific study context. 

Moreover, we consulted experts in the field to determine an appropriate threshold for interpreting the Likert scale responses.

the correct option here was strongly disagree. This was intentionally included to assess the perception of participants.

This is now explained in the methods section

 5

5-6

6 Line 113

Line 115-120

Line 121-123

Results

Socio-demographic characteristics

Line 135, The proportion 51.9% cannot be regarded as ‘about half’.

Table 1

Use Percent instead of Percentage

Include (n=317) for variables related to the spouses

Corrected 

Done. 

NB: Variables related to spouses is 337 not 317 6

7

 Line 136-137

Line 143-144

Discussion

Page 317, should be use of TBA only

Page 318; Those with secondary education and lower used TBA only compared with those with post secondary education. This statement should be reviewed.

 Done

Respondents’ level of education was significantly associated with the use of TBA only.

Reviewed. 

Those who had lower up to secondary education levels used only TBAs compared with those with post-secondary education for ANC. 18

18 Line 319

Line 319-321

Line 352; This study was community based with…….. This statement is not correct. The study is like a ‘health facility based’ study and not a community based study. The study took place among those assessing care among traditional birth attendants.

 Noted. The statement has been deleted 

Limitation

Delete the first statement under this sub-heading. Keep the second statement. Recall the 100% positive perception of the respondents as obtained in this study and state that it may be due to the study being conducted among those assessing care among TBAs. 

 DONE; Rephrased as

“Causation cannot be ascribed to the relationships found in this study. The 100% positive perception of the respondents obtained in this study may be due to the study being conducted among pregnant women assessing care with TBAs.” 

 20 Line 357-359

References

Authors should ensure that referencing style conforms to Journal guidelines.

 This is ensured. Thank you

---

## [Decision Letter · Decision Letter 3]

31 Jan 2024

PONE-D-23-19072R3Perception and reasons for choice of informal provider among women receiving antenatal care from traditional birth attendants in rural communities of Lagos state, NigeriaPLOS ONE

Dear Dr. Oluwole,

Thank you for submitting your manuscript to PLOS ONE. After careful consideration, we feel that it has merit but does not fully meet PLOS ONE’s publication criteria as it currently stands. Therefore, we invite you to submit a revised version of the manuscript that addresses the points raised during the review process.

We look forward to receiving your revised manuscript.

Kind regards,

Adetayo Olorunlana, Ph.D.

Academic Editor

PLOS ONE

Journal Requirements:

Reviewers' comments:

Reviewer's Responses to Questions

**Comments to the Author**

1. If the authors have adequately addressed your comments raised in a previous round of review and you feel that this manuscript is now acceptable for publication, you may indicate that here to bypass the “Comments to the Author” section, enter your conflict of interest statement in the “Confidential to Editor” section, and submit your "Accept" recommendation.

Reviewer #2: (No Response)

2. Is the manuscript technically sound, and do the data support the conclusions?

Reviewer #2: Yes

3. Has the statistical analysis been performed appropriately and rigorously? 

Reviewer #2: No

4. Have the authors made all data underlying the findings in their manuscript fully available?

Reviewer #2: Yes

5. Is the manuscript presented in an intelligible fashion and written in standard English?

Reviewer #2: Yes

6. Review Comments to the Author

Reviewer #2: Review comments

General comments

I applaud the efforts of Authors in reviewing this manuscript to this very stage. Well done. I may request for a minor review of the manuscript which should be the last in the review process.

Authors should take note of the following points;

1. Basis for inclusion of variables into the logistic regression model after bivariate analysis. This was not stated all through the manuscript but from the results of the study as presented, Authors only included variables that were statistically significant at p<0.05. This may not be the best approach. I encourage Authors to use a cut off p value of <0.2 for inclusion of variables into the regression model after bivariate analysis. Refer to

Maldonado G, Greenland S. Simulation study of confounder-selection strategies. American Journal of Epidemiology. 1993;138(11).

2. The presentation of the Abstract was not done according to Journal guidelines. Authors should refer to the website of the Journal for guidance. Also, positive perception and utilization being the outcome variables ought to be defined in the methods section of the abstract. Authors should also have the key words indicated.

3. In the Background section, the first definition of TBA should suffice.

4. Authors should indicate the date internet articles were accessed in the reference section. They should also ensure that use of references conform to Journal guidelines.

5. Instead of ‘None’ for education of spouse, Authors should use No formal education and sample size, (n=337) should be placed under the Frequency column for uniformity. For table 2, reporting of results should be based on proportions instead of raw figures.

6. Provide the rationale for using <70% of the total score as basis for categorizing a respondent as having good perception of TBAs.

7. PLOS authors have the option to publish the peer review history of their article (what does this mean?). If published, this will include your full peer review and any attached files.

Reviewer #2: **Yes: **EDMUND NDUDI OSSAI

---

## [Author Response · Author response to Decision Letter 3]

17 Feb 2024

Authors should take note of the following points;

1. Basis for inclusion of variables into the logistic regression model after bivariate analysis. This was not stated all through the manuscript but from the results of the study as presented, Authors only included variables that were statistically significant at p<0.05. This may not be the best approach. I encourage Authors to use a cut off p value of <0.2 for inclusion of variables into the regression model after bivariate analysis. Refer to

Maldonado G, Greenland S. Simulation study of confounder-selection strategies. American Journal of Epidemiology. 1993;138(11).

RESPONSE: We have now applied a cut-off p-value of <0.2 for the inclusion of variables into the regression model after bivariate analysis. (Pg. 6; Line 122-123)

The basis for the inclusion of variables into the logistic regression model after bivariate analysis have been stated. . (Pg. 6; Line 117-122)

2. The presentation of the Abstract was not done according to Journal guidelines. Authors should refer to the website of the Journal for guidance. 

RESPONSE: The abstract is now presented according to the journal guidelines with sections. (Pg. 2-3; Line 29-56)

Also, positive perception and utilization being the outcome variables ought to be defined in the methods section of the abstract. 

RESPONSE: 

Positive perception of TBA services corresponds with a score of <70%. Utilization of ANC services was dichotomized into the use of TBA alone or the use of a combination of TBA and HCFs for ANC services. This has been included in the manuscript. (Pg. 6; Line 130-132)

Authors should also have the keywords indicated.

RESPONSE: The keywords were included in the appropriate section of PLOS ONE Editorial Manager Submission system (ONLINE)

3. In the Background section, the first definition of TBA should suffice.

RESPONSE: Only the first definition of TBA is now maintained in the manuscript. (Pg. 3; Line 64-66)

4. Authors should indicate the date internet articles were accessed in the reference section. They should also ensure that use of references conform to Journal guidelines.

RESPONSE: DONE. (References) 

5. Instead of ‘None’ for education of spouse, Authors should use No formal education and sample size, (n=337) should be placed under the Frequency column for uniformity. 

RESPONSE: 

These corrections have been effected in the manuscript accordingly. (Pg. 8; Line 159-160)

For table 2, reporting of results should be based on proportions instead of raw figures.

These corrections have been effected in the manuscript accordingly. (Pgs. 8-9; Line 163-173)

6. Provide the rationale for using <70% of the total score as basis for categorizing a respondent as having good perception of TBAs.

RESPONSE: Theoretically, if the median score is used, respondents who have a neutral response for all items will be wrongly classified as having a negative perception (less supportive) of TBAs, hence, the need to increase the cut off point to 70%. 

---

## [Decision Letter · Decision Letter 4]

11 Mar 2024

PONE-D-23-19072R4Perception and reasons for choice of informal provider among women receiving antenatal care from traditional birth attendants in rural communities of Lagos state, NigeriaPLOS ONE

Dear Dr. Oluwole,

Thank you for submitting your manuscript to PLOS ONE. After careful consideration, we feel that it has merit but does not fully meet PLOS ONE’s publication criteria as it currently stands. Therefore, we invite you to submit a revised version of the manuscript that addresses the points raised during the review process.

We look forward to receiving your revised manuscript.

Kind regards,

Adetayo Olorunlana, Ph.D.

Academic Editor

PLOS ONE

Journal Requirements:

Reviewers' comments:

Reviewer's Responses to Questions

**Comments to the Author**

1. If the authors have adequately addressed your comments raised in a previous round of review and you feel that this manuscript is now acceptable for publication, you may indicate that here to bypass the “Comments to the Author” section, enter your conflict of interest statement in the “Confidential to Editor” section, and submit your "Accept" recommendation.

Reviewer #2: All comments have been addressed

2. Is the manuscript technically sound, and do the data support the conclusions?

Reviewer #2: Yes

3. Has the statistical analysis been performed appropriately and rigorously? 

Reviewer #2: No

4. Have the authors made all data underlying the findings in their manuscript fully available?

Reviewer #2: Yes

5. Is the manuscript presented in an intelligible fashion and written in standard English?

Reviewer #2: Yes

6. Review Comments to the Author

Reviewer #2: Review comments

General comment

I commend the Authors for the review of the manuscript so far. I was of the opinion that this review will be the last however there are some observations I have made which may necessitate another review of the manuscript by the Authors. I plead with the Authors to bear with me in this regard.

Authors should take note of the following points

1. Table 5: Predictors of health service utilization among the respondents. Authors included the educational level of the Spouse in the logistic regression model along with other variables. From table 1, there are 347 respondents of which 337 are living with their spouses. This means that the regression was done based on 337 which is the number common to all. This wrong. Authors should repeat the regression analysis without including the educational level of the Spouse. The inclusion of the educational level of the Spouse for further analysis should stop at the bivariate level.

2. Naming the outcome variable, ‘Use of health service utilization’ appear very confusing and may be misunderstood by Readers. The outcome measure should be utilizing only TBA services in current pregnancy. Authors are free to paraphrase the outcome measure as stated.

3. In the abstract, the results of the logistic regression analysis should be represented using adjusted odds ratio and 95% confidence interval. For example, Predictors of utilizing only TBA services in current pregnancy included …………

4. Define ‘positive perception’ in the methods section of abstract.

5. SPSS is currently referred to as Statistical Product and Service Solutions. Authors should reflect this correction in the manuscript.

6. Respondents should report the results of the bivariate analysis bearing in mind the difference in proportions. For example, ‘The highest proportion of respondents who utilized only TBA services, 80.9% had primary education while the least, 38.8% have attained tertiary education and the difference in proportions was found to be statistically significant, (χ2 =39.896, p<0.001)

7. Authors should present the tables in the manuscript in line with Journal guidelines. Authors should refer to the Journal website for guidance.

8. Authors should ‘expand’ the study limitation so as to enhance the understanding of Readers. The caption ‘Limitation’ may be deleted.

9. In table 3, use ‘estimated distance’ instead of ‘distance’. In Line 101, use ‘TBAs’. Line 141, use ‘name’ instead of ‘names’ Line 164, use ‘Less than half of the respondents, 43.6%.

10. Use ‘Percent’ instead of ‘Percentage’. p<0.001 instead of p=<0.001. In table 2 include the meaning of TBA as a foot note.

7. PLOS authors have the option to publish the peer review history of their article (what does this mean?). If published, this will include your full peer review and any attached files.

Reviewer #2: **Yes: **EDMUND NDUDI OSSAI

---

## [Author Response · Author response to Decision Letter 4]

30 Mar 2024

Response to Reviewers

PONE-D-23-19072R4 30-03-2024

Perception and reasons for the choice of informal provider among women receiving antenatal care services from traditional birth attendants in rural communities of Lagos state, Nigeria

Reviewer #2: Review comments

S/no Reviewers’ comments Authors’ comments Page Line Number

1. Table 5: Predictors of health service utilization among the respondents. Authors included the educational level of the Spouse in the logistic regression model along with other variables. From table 1, there are 347 respondents of which 337 are living with their spouses. This means that the regression was done based on 337 which is the number common to all. This wrong. Authors should repeat the regression analysis without including the educational level of the Spouse. The inclusion of the educational level of the Spouse for further analysis should stop at the bivariate level.

 We have repeated the regression analysis without including the educational level of the Spouse. The inclusion of the educational level of the Spouse for further analysis should has now stopped at the bivariate level. 17 258-261

2. Naming the outcome variable, ‘Use of health service utilization’ appear very confusing and may be misunderstood by Readers. The outcome measure should be utilizing only TBA services in current pregnancy. Authors are free to paraphrase the outcome measure as stated.

 The outcome measure has been rephrased as “utilizing only TBA services in current pregnancy”

Table 3: Utilization of TBA services only in current pregnancy among respondents

Adjustment has been made throughout the manuscript 12-13 215-216

3. In the abstract, the results of the logistic regression analysis should be represented using adjusted odds ratio and 95% confidence interval. For example, Predictors of utilizing only TBA services in current pregnancy included …………

 The results of the logistic regression analysis have been represented using adjusted odds ratio and 95% confidence interval in the abstract 3 54-56

4. Define ‘positive perception’ in the methods section of abstract. Defined in method section. 2 42-43

5. SPSS is currently referred to as Statistical Product and Service Solutions. Authors should reflect this correction in the manuscript. corrected in the manuscript 6 121

6. Respondents should report the results of the bivariate analysis bearing in mind the difference in proportions. For example, ‘The highest proportion of respondents who utilized only TBA services, 80.9% had primary education while the least, 38.8% have attained tertiary education and the difference in proportions was found to be statistically significant, (χ2 =39.896, p<0.001)

 Report of the results of the bivariate analysis has been reported in proportions. 13-14 219-234

7. Authors should present the tables in the manuscript in line with Journal guidelines. Authors should refer to the Journal website for guidance. Done. Each variables are now in separate cells according to the Journal guidelines 8-17 

8. Authors should ‘expand’ the study limitation so as to enhance the understanding of Readers. The caption ‘Limitation’ may be deleted.

 This has been done. The caption ‘Limitation’ is deleted. 23 398-402

9. In table 3, use ‘estimated distance’ instead of ‘distance’. In Line 101, use ‘TBAs’. Line 141, use ‘name’ instead of ‘names’ Line 164, use ‘Less than half of the respondents, 43.6%. All the corrections has been effected accordingly 

10. Use ‘Percent’ instead of ‘Percentage’. p<0.001 instead of p=<0.001. In table 2 include the meaning of TBA as a foot note. Done. meaning of TBA has been included as a foot note in table 2 10 

181-182

---

## [Decision Letter · Decision Letter 5]

17 Apr 2024

PONE-D-23-19072R5Perception and reasons for the choice of informal provider among women receiving antenatal care services from traditional birth attendants in rural communities of Lagos state, NigeriaPLOS ONE

Dear Dr. Oluwole,

Thank you for submitting your manuscript to PLOS ONE. After careful consideration, we feel that it has merit but does not fully meet PLOS ONE’s publication criteria as it currently stands. Therefore, we invite you to submit a revised version of the manuscript that addresses the points raised during the review process.

We look forward to receiving your revised manuscript.

Kind regards,

Adetayo Olorunlana, Ph.D.

Academic Editor

PLOS ONE

Journal Requirements:

Reviewers' comments:

Reviewer's Responses to Questions

**Comments to the Author**

1. If the authors have adequately addressed your comments raised in a previous round of review and you feel that this manuscript is now acceptable for publication, you may indicate that here to bypass the “Comments to the Author” section, enter your conflict of interest statement in the “Confidential to Editor” section, and submit your "Accept" recommendation.

Reviewer #2: All comments have been addressed

2. Is the manuscript technically sound, and do the data support the conclusions?

Reviewer #2: Yes

3. Has the statistical analysis been performed appropriately and rigorously? 

Reviewer #2: No

4. Have the authors made all data underlying the findings in their manuscript fully available?

Reviewer #2: Yes

5. Is the manuscript presented in an intelligible fashion and written in standard English?

Reviewer #2: Yes

6. Review Comments to the Author

Reviewer #2: Review comments

General comment

My intention was to approve the publication of this manuscript based on this review. However on close scrutiny, there is a flaw in the analysis and interpretation of study findings which will be too grave to ignore. This may necessitate that the manuscript may require another revision before it could be accepted for publication. I urge the Authors to critically review this manuscript before the next submission.

Authors should take note of the following observations.

1. Binary logistic regression result. In the abstract, Authors made it look like having a health insurance coverage predicted the use of TBA only in the study which is the outcome variable. See also Lines 248-250 in the result section and lines 388-390 in the discussion section.

Compare these comments with what was showed in table 4 where a significantly higher proportion of respondents who had no insurance coverage, 76.3% utilized only TBA services when compared with those who had insurance coverage, 41.7% (p<0.001). See also the result in table 5 on the same independent variable. I wonder why there was a re-arrangement of the categories of the independent variable ‘Having insurance coverage’ in tables 4 and 5.

IBM-SPSS by default utilizes the last category of the independent variable as the reference category in logistic regression analysis. The question will be whether the Authors coded the variable, ‘Respondent having insurance coverage’ in two different forms in the data set. This is why the results of Chi square and logistic regression analysis are sometimes included in the same table.

Also, the result as displayed for the educational attainment of the respondents in tables 4 and 5 seem not to move in tandem. This is because at bivariate analysis (Chi square), 58.3% of respondents who had no formal education utilized only TBA services as against 38.8% for those who have attained tertiary education. However, at logistic regression analysis, the respondents who have attained tertiary education were 3.480 times more likely to utilize TBA services only.

Authors should endeavor to clarify these two observations in the next submission and if possible seek the services of a Statistician.

2. Minor corrections. Line 41, use ‘included’ instead of ‘include’.

In line 52, use ‘perceived spiritual protection’. Line 55 should be having no health insurance coverage. Report the finding on educational level accordingly after the re-analysis.

In line 68, use TBAs. Use TBAs also in line 71 and 82.

Line 195 should also be perceived spiritual protection.

Line 231, use higher proportion since the categories of the independent variable are two in number. Delete least in line 233.

In table 5, use p<0.001 instead of 0.000.

Authors should remember to report findings with regards to health insurance and educational level at logistic regression analysis accordingly.

7. PLOS authors have the option to publish the peer review history of their article (what does this mean?). If published, this will include your full peer review and any attached files.

Reviewer #2: **Yes: **EDMUND NDUDI OSSAI

---

## [Author Response · Author response to Decision Letter 5]

12 May 2024

Response to Reviewers/Rebuttal letter

PONE-D-23-19072R5

 12-05-2024

Perception and reasons for the choice of informal provider among women receiving antenatal care services from traditional birth attendants in rural communities of Lagos state, Nigeria

Reviewer #2: Review comments

S/no Reviewers’ comments Authors’ comments Page Line Number

1. Binary logistic regression result. In the abstract, Authors made it look like having a health insurance coverage predicted the use of TBA only in the study which is the outcome variable. See also Lines 248-250 in the result section and lines 388-390 in the discussion section.

Compare these comments with what was showed in table 4 where a significantly higher proportion of respondents who had no insurance coverage, 76.3% utilized only TBA services when compared with those who had insurance coverage, 41.7% (p<0.001). See also the result in table 5 on the same independent variable. I wonder why there was a re-arrangement of the categories of the independent variable ‘Having insurance coverage’ in tables 4 and 5.

 Thank you so much sir, for taking note of this. It was a great mistake due to a re-arrangement of the categories of the independent variable. This has been corrected a new table 5 which was re-run by a statistician has been inserted with necessary adjustments accordingly. 15 241-242

2. 2. Minor corrections. Line 41, use ‘included’ instead of ‘include’.

In line 52, use ‘perceived spiritual protection’. 

In line 68, use TBAs. Use TBAs also in line 71 and 82.

Line 195 should also be perceived spiritual protection.

Line 231, use higher proportion since the categories of the independent variable are two in number. Delete least in line 233.

In table 5, use p<0.001 instead of 0.000.

 All has been effected in their respectives lines as indicated. As specified

3. Authors should remember to report findings with regards to health insurance and educational level at logistic regression analysis accordingly.

4. Reference 39 is OFF REMOVED AND DELETED BECAUSE TABLE 5 HAS BEEN REVISED. The reference 39 is no longer useful

---

## [Editor Report · Decision Letter 6]

21 May 2024

Perception and reasons for the choice of informal provider among women receiving antenatal care services from traditional birth attendants in rural communities of Lagos state, Nigeria

PONE-D-23-19072R6

Dear Dr. Oluwole,

We’re pleased to inform you that your manuscript has been judged scientifically suitable for publication and will be formally accepted for publication once it meets all outstanding technical requirements.

Kind regards,

Adetayo Olorunlana, Ph.D.

Academic Editor

PLOS ONE
---

## [Editor Report · Acceptance letter]

22 May 2024

PONE-D-23-19072R6 

PLOS ONE

Dear Dr. Oluwole, 

I'm pleased to inform you that your manuscript has been deemed suitable for publication in PLOS ONE. Congratulations! Your manuscript is now being handed over to our production team.

Kind regards, 

on behalf of

Associate Professor Adetayo Olorunlana 

Academic Editor

PLOS ONE